Hormonal and psychological influences on performance anxiety in adolescent female volleyball players: a multi-approach study

http://orcid.org/0000-0002-1208-1034 Rossi Carlo 1 2
Amato Alessandra 3
Alesi Marianna 4
Alioto Anna 1
Schiera Gabriella 5
Drid Patrik 6
Messina Giulia 1
Pagliaro Andrea 1
Di Liegro Italia 7
Proia Patrizia 1 patrizia.proia@unipa.it
1 Sport and Exercise Sciences Research Unit, University of Palermo , Palermo , Italy
2 Research and Innovation, Centro Medico di Fisioterapia “Villa Sarina” , Trapani , Italy
3 Department of Biomedical and Biotechnological Sciences, Section of Anatomy, Histology and Movement Science, School of Medicine, University of Catania , Catania , Italy
4 Department of Psychology, Educational Sciences and Human Movement, University of Palermo , Palermo , Italy
5 Department of Biological, Chemical and Pharmaceutical Sciences and Technologies (STEBICEF), University of Palermo , Palermo , Italy
6 Faculty of Sport and Physical Education, University of Novi Sad , Novi Sad , Serbia
7 Department of Biomedicine, Neurosciences and Advanced Diagnostics, University of Palermo , Palermo , Italy
Oliveira Sonia
Electronic publication date: 2024 Feb 19
Publication date: 2024
Volume: 12
Electronic Location ID: e16617
Received 2023 Jun 2; Accepted 2023 Nov 15
Copyright: © 2024 Rossi et al.
Copyright year: 2024
Copyright holder: Rossi et al.
License: This is an open access article distributed under the terms of the Creative Commons Attribution License, which permits unrestricted use, distribution, reproduction and adaptation in any medium and for any purpose provided that it is properly attributed. For attribution, the original author(s), title, publication source (PeerJ) and either DOI or URL of the article must be cited.
License URL: https://creativecommons.org/licenses/by/4.0/

Keywords: Performance anxiety, Cortisol, Amylase, Team sports, Volleyball, Biochemical parameters, Exercise

Funding: “Villa Sarina” Physiotherapy Medical Center, 91011 Alcamo, Italy This manuscript is supported by a doctoral scholarship granted by the "Villa Sarina" Physiotherapy Medical Center, 91011 Alcamo, Italy. The funders had no role in study design, data collection and analysis, decision to publish, or preparation of the manuscript.

==============================
Background

The neuroendocrine system has important implications for affiliation behavior among humans and can be used to assess the correlation between social relationships, stress, and health. This can be influenced by social closeness; this aspect is the closeness towards another individual or a group of individuals such as a sports team. Sports performance anxiety is considered an unpleasant emotional reaction composed of physiological, cognitive, affective, and behavioral components. This motivates us to learn about the process that can influence the outcome of competition. Hormones and genetics would seem to influence outcome and performance. In this regard, many studies have focused on the exercise response as a function of ovarian hormones and it has been observed that progesterone is a hormone that plays a key role in reducing anxiety, and thus stress, in humans and other animals. On the other hand, high cortisol concentrations are known to contribute to increased anxiety levels. However, the salivary alpha-amylase (sAA) enzyme has been suggested as marker of acute stress than cortisol. Genetics also seem to influence anxiety and stress management as in the case of brain-derived neurotrophic factor (BDNF) and striatal dopamine transporter (DAT). Therefore, the study aims to investigate social closeness, as a measure of sports team cohesion that can influence athletes’ performance results, and its ability to influence the secretion of hormones, such as progesterone and cortisol, that affect the management of sports anxiety while also taking into account genetic background during a volleyball match.

Methods

Twenty-six female volleyball players who volunteered participated in this study (mean ± SD: age, 12.07 ± 0.7 years), and played in the final of the provincial volleyball championship in Palermo. All girls were during the ovarian cycle, in detail between the follicular and early ovulatory phases.

Results

The results showed a significant decrease in salivary cortisol only in the winning group (p < 0.039). In fact, whilst in the latter the pre-match level was 7.7 ng/ml and then decreased to 4.5 ng/ml after the match, in the losers group change was not statistically significant (7.8 ng/ml vs 6.6 ng/ml pre- and post-match). As to the sAA concentration, the winning team showed a statistically significant variation between pre- and post-match than the losers (166.01 ± 250 U/ml vs 291.59 ± 241 U/ml) (p = 0.01).

Conclusion

Analyzing the results of the SAS-2 psychological test it is highlighted that, on average, the loser group was more anxious than the winning group, and this contributed to the final result. In conclusion, there is strong evidence supporting the state of the art that many factors can affect performance anxiety and thus the performance itself.

Introduction

The neuroendocrine system has important implications to understand social bonds, and affiliate behavior among humans and to evaluate the correlation between social relations, stress, and health. One factor that influences this, is the closeness and the athlete’s sports team membership (Schubert & Otten, 2002).

In addition, the study of individualism, defined as the assertion of self-interest and success, is also essential to know whether the individual acts to make his or her interest override the interest of the group. In order to detect this status, the main instrument used is the Auckland Scale for Individualism and Collectivism (AICS) (Shulruf, Hattie & Dixon, 2007). Another factor is performance anxiety usually assessed by the Sport Anxiety Scale-2 (SAS-2) developed by Smith et al. (2006); it measures three components of anxiety: somatic, worry and concentration disruption that are more involved in sports competition (Fuentes-Rodriguez, Garcia-Lopez & Garcia-Trujillo, 2018). Recent data suggest that progesterone shows the reasons why a person builds relationships with others (Brown et al., 2009). It is a steroid hormone synthesized from cholesterol in the ovaries and adrenal cortex; after production by the corpus luteum, it inhibits the release of follicle-stimulating hormone (FSH) and luteinizing hormone (LH), along with estrogen, through a negative feedback mechanism on the hypothalamus. It has been noticed that progesterone has an important role in stress, and in particular, it reduces anxiety both in humans and in animals through its metabolite, allopregnanolone (Li & Graham, 2017). The stress decrease due to progesterone may be both a prerequisite and a consequence of membership in a group, social cohesion, and the ensuing altruistic behavior and social closeness.

Some studies have focused on the exercise response to ovarian hormones (Bonen et al., 1981; Jurkowski et al., 1978). A recent study shows that LH and FSH were significantly increased immediately after exercise particularly starting with running (8 km/h) increasing 0.5 km/h per minute, instead, progesterone activity decreased significantly (Otağ et al., 2016). Progesterone is a hormone detectable in saliva as well as other hormones like cortisol, androgens (including testosterone and dehydroepiandrosterone), estrogen, and aldosterone. Recent studies show that progesterone levels significantly increase from menstruation (M) to luteal phase (L). In particular, Żelaźniewicz et al. (2016) stated that the levels of progesterone in a sample of 50 women (ranged from 19.6 to 36.1 years) during menstruation, in which it was found to an average of (0.3 ng/ml), whilst during mid-luteal phase (L), the levels increased up to an average of (17.7 ng/ml). Several working groups are conducting research in the field of stress marker detection to develop reliable, rapid, and accessible methods. Many stress-related biomarkers are currently under investigation (Giacomello, Scholten & Parr, 2020). These include some of the following.

Cortisol is a steroid hormone that belongs to the glucocorticoid family. It is secreted by the hypothalamic-pituitary-adrenal (HPA) axis and increases in response to stressful stimuli: for instance, anxiety, depression, and intense physical exercise (Von Dawans et al., 2018). Cortisol is known as the main responsible for catabolic pathways, since it reduces protein synthesis, increases protein degradation, and inhibits the inflammatory and immune process; about exercise, the release of hormones also depends on the workout duration and intensity. High-intensity exercise, ≥60% of the maximum rate of oxygen consumption (VO2 max) for at least 20–30 min, may cause several changes in cortisol levels (Stajer, Vranes & Ostojic, 2020). Saliva is a useful tool to evaluate physiological biomarkers above all in sports; it is used for monitoring steroids, peptides, and immune markers allowing it to become very popular in many types of research. The saliva can be collected quickly, and frequently and is a non-invasive method; it requires no medical education and can be performed directly on the sports field (Cannizzaro, Proia & Contro, 2015; Kreusser et al., 1972). Saliva is also rich in organic constituents, which include proteins, albumin, urea, uric acid, lactate, creatinine, insulin, and mucosal immunity markers such as immunoglobulins (IgA, IgM, and IgG), salivary alpha-amylase (sAA), lysozyme and lactoferrin (Engels et al., 2018). As regards sAA, a marker of the Sympatho Adrenal Medullary (SAM) activity through adrenergic receptors is considered to be a stress marker (Granger et al., 2007). Short-lived modifications in an increase of the salivary concentrations of sAA have been reported during competitions (Piacentini et al., 2015) under physically and psychologically stressful conditions (Kivlighan & Granger, 2006). Engert et al. (2011) highlighted the correlation between sAA levels and anxiety. A recent study shows how sAA could be considered a better indicator of acute stress than cortisol by stating that sAA is sensitive to both physical and psychological stress (De Pero et al., 2015). Another study emphasized the positive correlation between the sAA concentrations and the anxiety level of the pre-competition state; in fact, during this time the elite athletes are highly activated to react to mental and physical demands (Piacentini et al., 2015). This can affect the performance and indirectly the outcome.

The autonomic nervous system, controls saliva’s flow and composition, volume, and ionic/protein profile content. Particularly, α or β-adrenergic stimulation may change the amount, viscosity, protein concentration, and the saliva’s ion concentration.

Recently, social network analysis (SNA) models have been employed to explain the association between the chemical signals of the hypothalamic-pituitary-adrenal (HPA, e.g., cortisol) and social context elements, such as competition, social threat or social anxiety (Eisenegger, Haushofer & Fehr, 2011; Kornienko et al., 2014). A higher rate of affiliation, motivation, or social closeness induces an increase in progesterone levels and saliva flow because the positive effects come out from the sense of belonging and the interactions with others (Brown et al., 2009).

Moreover, higher cortisol rates are being widely demonstrated to be related to anxiety. This is a discomfort reaction, to which contributes physiological, cognitive, affective, and behavioral components, associated with unpleasant emotions and a state of distress occurring in response to events perceived as threatening to self-esteem (Krohne & Hock, 2011; Rappo, Alesi & Pepi, 2017).

Sport is an important test bank where the athlete’s value is mapped out and affects the process of elaborating self-esteem and self-efficacy. Athletes often experience unpleasant emotions and a state of anxiety induced by sports competitions. In other words, putting together this preliminary knowledge, it is possible to hypothesize that performance anxiety influences competition performance, leading to an increase in stress-related biomarkers such as cortisol and alpha-amylase. Furthermore, an increased sense of team affiliation and interactions with others could induce an increase in progesterone levels and salivary flow and this could be correlated with an anxiety reduction. Therefore, in this case, it would be possible to assume that high social closeness in the winning team, as well as high progesterone levels associated with low cortisol and alpha-amylase levels, would be considered positive markers for performance.

Also, genetics can play an important role in the onset of anxiety and stress management (Ising & Holsboer, 2006); in particular, the Brain-Derived Neurotrophic Factor (BDNF) that affects dopaminergic and serotonergic neuronal systems and plasticity and Dopamine Transporter (DAT), one of the main proteins in the dopaminergic system seem to influence emotions (Michałowska-Sawczyn et al., 2020). The BDNF gene is located in a region of chromosome 11 and several polymorphisms have been discovered within it, including the val66met polymorphism (G/A polymorphism) that is also correlated with the psychological response to stress and the motivation of physical exercise (Tsuru et al., 2014). There are few studies evaluating the link between the Val66Met polymorphism and physical activity response; most report no association (Canivet et al., 2015; Erickson et al., 2013; Flack et al., 2019; Watts, Andrews & Anstey, 2018; Zarza-Rebollo et al., 2022), while one study reported that Val/Val genotype experienced greater exertion compare with Met/Met genotype (Bryan et al., 2007). The literature has also dealt with intrinsic motivation during exercise in regular athletes, finding greater intrinsic motivation in Met carriers compared to Val/Val participants (Caldwell Hooper, Bryan & Hagger, 2014). This suggests that there may be an effect mediated by genetic factors. DAT plays a critical role in re-uptaking dopamine in presynaptic neurons (McHugh & Buckley, 2015). It is codified from the SLC6A3 gene, mapped on the short arm of chromosome 5; many studies highlighted the correlation between some DAT’s gene variants and the correlation with motivation and human behavior. It was observed that anxiety-related genotypic variants of DAT1, in particular the VNTR 9/10 variant, lead to lower levels of anxiety in athletes (Michałowska-Sawczyn et al., 2020). In contrast, there is a statistically significant increase for the group of athletes carrying the 10/10 genotype (Burke et al., 2011; Zarza-Rebollo et al., 2022).

In the available scientific literature, little is known about the correlation between social closeness, genetic background, and the level of some salivary hormones and performance anxiety; therefore, our study aimed to find possible relationships between these variables in a group of adolescent female volleyball players.

Materials and Methods

Participants

Twenty-six female volleyball players (mean ± SD: age, 12.07 ± 0.7 years), who took part in a provincial final match, volunteered to participate in this study. All participants were between the follicular and early ovulatory phase of the ovarian cycle. The psychometric and biological assessments were performed before and after the final match of the provincial volleyball championship in Palermo. As a consequence of the outcome of the match, they were randomly assigned to two groups: losers and winners. Participants provided informed consent before the study commenced and the procedures were performed following the Declaration of Helsinki. We subsequently received written informed consent from your study participants. The study was approved by the ethics committee of the University of Novi Sad, Serbia (ref. no. 46-06-02/2020-2).

Psychometric assessments

Sport anxiety scale-2 (SAS-2)

The Sport Anxiety Scale-2 (SAS-2) developed by Smith et al. (2006) consisted of 15 items aimed at measuring the three components of anxiety: somatic, worry, and concentration disruption. Firstly, five somatic items targeting stomach and muscle states indicating autonomic arousal; for example, “I feel tense in my stomach”. Secondly, worry items expressed behaviors revealing concerns about negative sports performance; for example, “I worry that I won’t play well”. Thirdly, the concentration disruption items stated difficulties in personal concentration causing task-irrelevant thoughts; for example, “It is hard to concentrate on the game, it is hard to concentrate on the game”. Participants were asked to read each item carefully and express the extent to which they experienced that behavior or internal state using a four-point answer scale ranging from 1 (not at all) to 4 (very much). The SAS-2 was used to assess self-reported anxiety.

Closeness

A measure of closeness to a sports team was developed for this study to measure the athlete’s perception of closeness and belonging to their own sports team. It was inspired by the IOS Overlap of Self, Ingroup, and Outgroup (Schubert & Otten, 2002) aims at measuring the personal perception of the self in the intergroup context. In detail, our measure was based on spatial metaphor evaluation; each athlete was given a white sheet with seven colored figures picturing seven levels of closeness between the self (a blue circle) and the own sports team (a green circle). He was asked to choose the picture better representing his perception of belonging to the team. The scoring was from 1 to 7 points with seven indicating a higher level of closeness.

Individualism/Collectivism

Participants were given a reduced version of the Auckland Individualism and Collectivism Scale (AICS) developed by Shulruf, Hattie & Dixon (2007) and adapted into the Italian context by Alesi and Pepi (Shulruf et al., 2011). This version consisted of eight items, four addressing collectivist attributes such as advice and harmony and four addressing individualistic attributes such as competition and uniqueness (for example “Before I make a major decision, I seek advice from people close to me”). Subjects were asked to read each sentence carefully and express the frequency from never or rarely to always. The scoring parameters were from 1 point to 6 points for the item.

Saliva collection and genotyping

Since the food can influence the salivary hormone concentrations, the subjects consumed the meal at least 90 min before taking the specimen. When the athletes got to the sports ground, the first saliva sample was taken by passive smearing (Pre), whilst the second sample was collected within 10 min after the end of the match (Post). The match was played between 15.00 and 18.00 and the samples were collected in a sterile 15 ml centrifuge tube and stored at −80 °C until assay. Each sample was analyzed twice in the same assay run. Both sAA and cortisol were employed as markers of stress.

The minimum detection limit for the cortisol assay was 0.12 ng/mL and for the progesterone assay 3.27 pg/mL.

The data was collected as previously described in Giustiniani et al. (2021), specifically genomic DNA was isolated from 1 ml of whole saliva using a PureLink kit (PureLink Genomic DNA; Thermo Fisher Scientific, Waltham, MA, USA) according to the manufacturer’s protocol. The genotyping was carried out by polymerase chain reaction (PCR) in a total reaction volume of 50 µl containing 50 ng of template, 1 µl of 10 mM deoxynucleotide triphosphate (dNTPs), 1 µl of 30 pmol each primer, and 5 µl of 10X reaction buffer with MgCl2. The target sequence was augmented using a 5U/µl Dream Taq (Thermo Fisher Scientific, Waltham, MA, USA) with the following primers: P1 (forward) 5′CCTACAGTTCCACCAGGTGAGAAGAGTG-3′; P2 (reverse) 5′ TCATGGACATGTTTGCAGCATCTAGGTA 3′; P3 (G allele specific-reverse) 5′ CTGGTCCTCATCCAACAGCTCTTCTATaAC 3′; P4 (A allele specific-forward) 5′ATCATTGGCTGACACTTTCGAACcCA 3′ used for determination of BDNF genotype and 5′ TGTGGTGTAGGGAACGGCCTGAG 3′ (forward) and 5′ GTTCCTGGAGGTCACGGCTCAAG 3′ (reverse) used to determine the DAT1 genotype.

PCR amplification was performed with the following protocol: denaturation at 94 °C for 5 min, followed by 35 cycles of denaturation at 94 °C for 30 s, annealing at 62.5 °C for 60 s for BDNF and 66 °C for 30 s for DAT1, extension at 72 °C for 30 s and final extension at 72 °C 5 min. The fragments were separated on 8% vertical polyacrylamide gel at 100 V for 1 h and visualized with ethidium bromide (Giustiniani et al., 2021). Genetic polymorphism analysis of BDNF and DAT1 genes was used to investigate the possible association between genetic mutations and stress response.

Salivary flow assessment

Unstimulated saliva was collected in a vial until the center line of the vial (5 ml) was reached. As it is possible that there will be a lot of foam, it will be necessary to wait a few minutes by placing the sample on ice, to wait for it to disappear and evaluate the amount of mL collected (Nagy et al., 2015). The saliva flow rate was calculated by dividing the amount of saliva (mL) collected by the time taken (minutes).

Amylase determination

Salivary alpha-amylase concentration was measured by the kinetic method at 405 nm and the activity was determined specifically using substrate 2-chloro-p-nitrophenyl-α-D-maltotrioside (CNP-G3) following the manufacturer’s instructions (Sigma Aldrich, St. Louis, MO, USA).

Enzyme-linked immunosorbent assay

Concentrations of saliva cortisol and progesterone were measured using 10 µl of a sample using human ELISA kits (DiaMetra S.r.l.; Segrate, Milan, Italy) according to the manufacturer’s protocols.

Statistical analyses

The statistical analyses were performed using the Statistical Package for the Social Sciences 23.0 (IBM SPSS STATISTICS 23, Armonk, NY, USA). Descriptive statistics are expressed as the mean and standard deviations for all variables. The normality distribution of data was checked with The Shapiro-Wilks test. A paired t-test analysis was applied to compare the mean of variables before and after the match in case of normality; conversely, the Wilcoxon test was used. Pearson’s correlation test was performed to evaluate the association between psychological and hormonal parameters. The significance level was set at p ≤ 0.05.

Results

From the initial twenty-six players enrolled in the study, only data from twenty players were retained for analysis based on their participation in the match.

Sports performance anxiety levels and social closeness assessment

In Table 1 we analyzed the results of the SAS-2 psychological test, concluding that, on average, the losing group was more worried than the winning group, and the high score in test shows (evaluate Somatic Anxiety, Worry, and Concentration Disruption) that is part of the performance anxiety (10.5 winners vs 12.3 losers). No significant difference was found for the different measures of Cognitive anxiety when the pooled data were compared across the two groups.

Table 1 Means ± SD—psychological parameters in winners and losers.

	Winners	Losers	p value	
Performance anxiety	26.2 ± 5.4	30.1 ± 5.6	0.13	
Closeness	6.8 ± 0.4	5.6 ± 1	0.008*	
Notes:

* Significant differences at the 0.01 level.

Concerning the Team relationship, the score obtained was identical within the whole winning group, while was inconstant and low, on average, in the loser group. According to the outcome, significant differences between winners and losers were observed and were significantly different (p = 0.008).

Salivary hormonal profile before and after the match

Even if there was great variability in the progesterone level within the groups, as shown in Table 2, the average of the value was comparable: no significant difference was detected before and after the match in both groups. Noteworthy, Pearson’s correlation shows a negative significant correlation between progesterone levels post-match and the relationship with team (social closeness) in losers (p < 0.05).

Table 2 Means ± SD—pre- and post-match levels of cortisol and progesterone in winners and losers groups.

	Winners		Shapiro
Wilks	Losers		Shapiro
Wilks	
	Pre	Post	p Value	Pre	Post	p value	
Progesterone (pg/ml)	266.9 ± 136	244.5 ± 76	0.46	<0.001#	266 ± 72	311 ± 181	0.22	0.065	
Cortisol (ng/ml)	7.7 ± 1.1	4.5 ± 2.2	0.039*	0.108	7.8 ± 0.7	6.6 ± 2.2	0.42	0.885	
Notes:

* Significant differences at the 0.05 level.

# Shapiro-Wilks significance 0.001.

Analyzing the pre-match levels, it was equivalent in both groups, but it is interesting to note that in the winning group, there was a direct quadratic relationship compared to the loser’s group in which there was an inverse quadratic relationship between pre- and post-match levels. As shown in Table 2, a significant decrease in salivary cortisol, from pre- to post-match levels (7.7 ng/mL vs 4.5 ng/mL), was observed in the winning group (p < 0.039). Conversely, in the loser group, the value increased in the pre-match period approximately to the winner’s value, but the change was not statistically significant compared to the post-match period value reached (7.8 ng/ml vs 6.6 ng/ml).

Salivary parameters

Salivary flow rate

The match did not significantly affect the salivary flow rate; there was just a light increase only in losers flow (0.38 ml/min vs 0.43 ml/min) with a p = 0.44.

Salivary a-amylase concentration

As shown in Table 3, a significant increase in sAA, from pre- to post-match levels, was observed in the winning group (166.01 ± 250 U/mL vs 291.59 ± 241 U/mL; p = 0.01).

Table 3 Means ± SD—pre- and post-match in winners and losers groups of salivary parameters.

	Winners		Shapiro Wilks	Losers		Shapiro Wilks	
	Pre	Post	p value	Pre	Post	p value	
Salivary flow rate (ml/min)	0.67 ± 0.35	0.65 ± 0.50	0.88	0.096	0.38 ± 0.21	0.43 ± 0.31	0.44	0.084	
Salivary alfa-amy concentration (U/ml)	166.01 ± 250	291.59 ± 241	0.01**	0.102	216.1 ± 142	259.5 ± 240	0.34	0.216	
Salivary alfa-amy secretion (U/min)	77.21 ± 62.8	197.64 ± 229.7	0.005**	<0.001#	62.7 ± 25.3	78.83 ± 36.8	0.14	0.271	
Notes:

** Significant differences at the 0.01 level.

# Shapiro-Wilks significance 0.001.

Salivary a-amylase secretion

Winners showed a higher sAA secretion than losers; starting from a comparable value in both groups (77.21 U/min in the winner’s group vs 62.7 U/min in losers) after the match there was an increase only in winners (p = 0.07) that is not explained by a rise of salivary flow rates and was not statistically significant.

BDNF and DAT1 genotyping and response to the effort test

We investigated whether the BDNF Val66Met polymorphism and the variable number tandem repeat (VNTR) at the 3′ end of the DAT1 gene affect the emotional response and the final score (Fig. 1).

Figure 1 BDNF Val66Met polymorphism and the variable number tandem repeat (VNTR) at the 3′ end of the DAT1 gene affect the emotional response and the final score.

(A) Genotyping of BDNF by tetra primer amplified mutation system-PCR resolved on a 6% polyacrylamide gel. Lane 1 val/met genotype, lanes 2–4 val/val genotype, lanes 5 met/met genotype. A total of 401 bp band represents the control amplicon, while the specific bands for the G and A alleles are depicted by the 253 and 201 bp amplicons. (B) DAT 1 PCR products resolved on a 2.5% agarose gel; the fragments obtained were 480 and 440 bp corresponding to the genotype with 10 and 9 repeats (R), respectively.

BDNF allele frequencies in the whole group were 20% for the A (Met) allele and 80% for the G (Val) allele, and BDNF genotype frequencies were 5% for AA (Met/Met), 30% for AG (Val/Met) and 65% for GG (Val/Val) genotypes. Noteworthy, GG genotypes it was more expressed in the winning group (72.7% vs 55.6%), whilst in the loser group AG genotypes were predominant (44.4% vs 18.2%) but not in a statistically significant way. Regarding DAT1 VNTR allele frequencies in the whole group were 37.5% for nine-repeat and 62.5% for 10-repeat; DAT1 VNTR genotype frequencies were 20% for 9/9, 35% for 9/10, and 45% for 10/10 genotypes. In the two groups, the genotype distributions were homogeneous in the loser group (33.3% for all genotypes) and heterogeneous in the winning group with a prevalence of the genotype 10.10 (54.5%). This difference was not statistically significant. All genotype distributions agreed with the Hardy-Weinberg equilibrium.

Discussion

The purpose of this study was to investigate the influence of psychological components, genetic background, and salivary hormone changes on performance anxiety during a volleyball match in a group of adolescent female players. Psychometric evaluations conducted to assess levels of sports performance anxiety through the SAS-2 test showed that the losing group was more worried than the winning group. This is evidenced by the high score on the SAS-2 test.

As for social closeness assessed through the IOS and AICS test, the score obtained was constant in the winning group, while it was inconsistent and lower in the losing group. A statistically significant difference was shown between the groups, showing a greater sense of belonging in the team of the winning group.

The results showed that performance outcome is independent of genetic background. The genotyping of the samples for BDNF and DAT1 polymorphisms did not highlight any correlation with the score of the match; nevertheless, the existing literature is controversial. For example, Grzywacz et al. (2021) found that DAT1 appeared to negatively affect the sports performance of 100 Polish athletes in mixed martial arts, Judo, boxing, karate, kickboxing, and wrestling. However, the nature of the sport, individual or team, may have a different influence on the factors described (Grzywacz et al., 2021). A systematic review in 2021 by Donati et al. (2021) defines BDNF as the main neurotrophin involved in sports performance. However, one study suggests that BDNF is definitely involved in physical activity, but current findings on mechanisms that might be involved in responses during physical exercise are not yet clear (Moreira et al., 2018). Furthermore, Bamaç et al. (2011) found that soccer can cause an increase in serum BDNF concentrations.

We found a significantly higher secretion of alpha-amylase in the winning team than in the losers, in disagreement with the existing literature that considers sAA as a factor that negatively affects performance (Heydari et al., 2022). However, our data support the hypothesis of several studies (Dehghan et al., 2019; Foretic et al., 2020; Kiely et al., 2019; Lim, 2016) that have demonstrated a positive correlation between anxiety and sAA levels, which increase when dealing with stressful situations, e.g., sports competitions. However, Lim (2016) showed there was a positive correlation between stress and sAA levels since as the time of competition approached, the stress increased the salivary hormone level; this is in agreement with the results of our study.

The natural ovarian cycle is characterized by large fluctuations in sex hormone concentrations that could affect athletic performance (Lebrun, 1993). Despite the large variability of progesterone in both groups, we found no statistically significant differences before and after the match. According to Brown et al. (2009) who investigated progesterone and its correlation with motivation, we can state that progesterone is probably randomly related to motivation during competition. However, it may be a stress marker response activation (Paul & Purdy, 1992). Anyway, some studies have shown links between progesterone and a higher motivational affiliation (Schultheiss, Wirth & Stanton, 2004; Wirth & Schultheiss, 2006). Regarding the link between cortisol levels and the result, we found that although the level was similar in both teams before the competition the losing group showed a higher cortisol level. Consistent with our results, Lautenbach et al. (2015) identify an association between cortisol and performance, confirming the relationship between a high cortisol level and poor performance. Our study agrees with Cameron et al. (2017) who used an adapted standardized test in adolescents, carrying out oral tasks and questionnaires, and confirmed that cortisol is a stress response. Lautenbach et al. (2015) evaluated cortisol levels in two male tennis players confirming that cortisol levels are higher in the losers of a competition. We can also state that the match can be interpreted as a stress factor, considering the higher levels of anxiety in the team that lost the competition (Filaire et al., 2009); for sure anxiety affects not only the subjects involved in sports teams but also in individual sports as demonstrated from Rossi et al. (2022), in a study on judo athletes in which the anxiety negative influence performance. Hence, competition preparation itself may produce a stress-enhancing physiological response (Ehrlenspiel & Strahler, 2012), and restoration of post-competition homeostasis requires significant mental effort (Kraemer et al., 2001).

This multidisciplinary approach was focused on better understanding the psycho-biological basis of anxiety performance. The hormonal change and psychological test results supported the scientific starting points on which we based our research design. Our data shows that the losing group was more anxious than the winning group and that they also had a high cortisol response. The trend in cortisol levels, on the other hand, shows a level that remains elevated only in the losing group, while it decreases in the winning group. As for sAA, it increases in both groups between pre- and post-race, but much more in the winning group where a statistically significant change is detected. So, ultimately, we have two putative markers of stress that act differently and it is not necessarily that sAA is not a better stress marker than salivary cortisol of stress.

Furthermore, the idea of correlating the activity of sAA to that of the sympathetic nervous system has provided interesting results for future investigations given that adrenergic stimulation alone during stress is only one of these factors.

Conclusions

Coaches play a crucial role in the motivation and emotions experienced by young athletes. However, another key factor is performance anxiety, which can be defined as a personality trait that causes a state of excessive emotional activation both before and during competition. The consequences of performance anxiety are highly disabling, as they affect sports performance, severely impairing concentration.

The knowledge gained through this pilot study could be a useful tool for stress management during competition. However, since there is no definitive proof that one or more of the parameters analyzed is the cause of winning and/or losing, it is possible to speculate that some of them such as cortisol and progesterone ‘may’ exert an influence on performance.

The preliminary results obtained need to be confirmed by increasing the sample size and including athletes of both sexes and practicing both individual and team sports.

Supplemental Information

Supplemental Information 1 Dataset of hormonal, psychological and genetic parameters.

The data shows all the values taken into account in the article in two different colours: yellow for losers (P) and green for winners (V). However, the Pre and Post measurements are specified.

Additional Information and Declarations

Competing Interests

Author Contributions

Human Ethics

Data Availability

The authors declare that they have no competing interests.

Carlo Rossi conceived and designed the experiments, prepared figures and/or tables, and approved the final draft.

Alessandra Amato analyzed the data, authored or reviewed drafts of the article, and approved the final draft.

Marianna Alesi performed the experiments, authored or reviewed drafts of the article, and approved the final draft.

Anna Alioto analyzed the data, prepared figures and/or tables, and approved the final draft.

Gabriella Schiera performed the experiments, authored or reviewed drafts of the article, and approved the final draft.

Patrik Drid performed the experiments, authored or reviewed drafts of the article, and approved the final draft.

Giulia Messina analyzed the data, prepared figures and/or tables, and approved the final draft.

Andrea Pagliaro analyzed the data, prepared figures and/or tables, and approved the final draft.

Italia Di Liegro conceived and designed the experiments, prepared figures and/or tables, authored or reviewed drafts of the article, and approved the final draft.

Patrizia Proia conceived and designed the experiments, authored or reviewed drafts of the article, and approved the final draft.

The following information was supplied relating to ethical approvals (i.e., approving body and any reference numbers):

The Ethics Committee of Commission University of Novi Sad Faculty of Sport and Physical Education has approved to carry out the study within its structures (Ethical Application Ref: 46-06-02/2020-2).

The following information was supplied regarding data availability:

The raw data are available as a Supplemental File.

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
