# Peer review of "Hormonal and psychological influences on performance anxiety in adolescent female volleyball players: a multi-approach study"

_PeerJ, doi:10.7717/peerj.16617_

## Round 0.1 · original submission · Major Revisions

Dear authors, thank you for your submission. Performance anxiety is a complex issue influenced by multiple factors, including psychological, environmental, and social elements. Research investigating the specific hormonal and genetic influences on performance anxiety in adolescent female volleyball players could help deepen our understanding of this phenomenon and may contribute to tailored interventions or support strategies. I acknowledge the magnitude of this research effort. At this moment, I believe that the manuscript should benefit from MAJOR revisions. Please, refer to the reviewers' feedback.

·

Basic reporting

Firstly, I would like to commend the authors on submitting this piece of work. Tackling a concept such as performance anxiety is an ambitious feat, and the level of research, attention to detail, and due diligence required to merely comprehend this concept has not gone unnoticed. The following comments have been written with the intention of helping the authors successfully publish their paper. Please take any critique as constructive, it is not from a place of condescension.

The paper struggles to conform to professional standards of courtesy and expression. There are many instances where the use of English could be improved to ensure that a global audience can easily understand the message being conveyed. Within the abstract alone, such instances are found in lines 36, 42, 46, 48, 50, 53, 55, 58 and 60. For context, the abstract is contained within lines 34 to 62. To best address this issue, I feel that the inclusion of colleagues who have a firm grasp of written English, or the use of a professional editing service, would greatly benefit the readability of this paper.

The introduction provides appropriate context, with the concepts of social relations and closeness elaborated upon – taking a sentence from this section and placing into the abstract would help the reader to understand the aim of the study when glancing through the abstract (see general comments - abstract). The references used to support the introduction are pertinent to the subject matter. The structure of the paper conforms to the Journal’s standards – the division of the piece into main headings and subheadings makes for an easier reading experience. All tables are relevant, well labelled, and appropriately referenced.

Experimental design

The topic at hand appears to be relatively novel and is well within the scope of the journal – which is fantastic. A research question is offered. However, a more structured “Problem, Gap, Hook” approach would improve the clarity of the research question and reader understanding (see general comments – introduction). The investigations detailed in the paper have clearly been performed well and satisfy any technical and ethical standards – ethics approval having been granted by the ethics committee of the Novi Sad University, Serbia. Methods are described with satisfactory detail and information to replicate (see general comments – materials and methods).

Validity of the findings

Reporting of results is satisfactory and reasonably easy to follow. Conclusions are adequate and related to supporting results. However, they may be a little confident (see general comments – conclusions).

Additional comments

Abstract
Whilst the abstract is relatively succinct, there are several statements that merit further contemplation. The first example of such a statement is: “Sports performance anxiety is a behavioral attitude that affects the outcome of athletes and teams.” The authors may wish to consider the concept of anxiety in terms of being classified as an attitude. It can often be the case that an individual (or indeed, group) is described as adopting a specific attitude to a specific event. Thus, can anxiety (or the subcategory of performance anxiety), which is a condition featuring in the International Classification of Diseases-11 (ICD-11), be a behavioural attitude? An excellent article the authors may wish to familarise themselves with is “Exploring the role of the DSM-5 performance-only specifier in adolescents with social anxiety disorder” by Fuentes-Rodrigue et al. (2018).

The second example is found in the latter half of the first statement. Namely, “…attitude that affects the outcome of athletes and teams.” I can understand why the authors may want to use such persuasive language; however, the beauty of science is that true causation is rarely found. In this case, is it entirely accurate to state that performance anxiety affects the outcome of athletes and teams? Or would it be better practice to state that performance anxiety “…can affect the outcome…”
Another example is the statement: “…progesterone is a hormone that plays a key role in reducing anxiety, and thus stress, in humans and other animals.” The authors must be careful not to conflate the terms anxiety and stress, as this can create additional confusion around two already nebulous terms. Is there any evidence to show that a reduction in anxiety, by subjective or objective measures, acts to reduce stress? If so, what type of stress are the authors referring to – emotional, psychological, or physical?

The aim of the study within the abstract is relatively clear. However, the background explaining why this study needs to be conducted doesn’t lead the reader to a logical conclusion that this is the best approach. For example, the abstract discusses performance anxiety, progesterone, cortisol, salivary alpha-amylase, brain-derived neurotrophic factor, striatal dopamine transporters, and then concludes that the objective is to investigate social closeness. It would greatly benefit the abstract (and the reader) if the concept of social closeness was explained.
Within the results section of the abstract, the authors refer to salivary cortisol data concerning the losing team, stating that “…the level remained almost unchanged (7.8 ng/ml vs. 6.6 ng/ml pre- and post-match). Whilst this may not constitute statistical significance, one can certainly say that an absolute difference of 1.2 ng/ml (≈ 15 % of the pre-match value) is not “almost unchanged.”

Introduction
The introduction serves its purpose as a piece of writing to give the reader appropriate background on the topic, as well as conveying the purpose of the study. It would be great however, to see more of the “Problem, Gap, Hook” approach. Namely, what is the actual problem, how are the team tackling this issue in a novel way, and what implications will this combination of techniques and results have for the field at large? The authors state that, “In the available scientific literature, little is known about the correlation between social closeness, genetic background and the level of some salivary hormones and performance anxiety; therefore, our study aimed to find possible relationships between these variables in a group of adolescent female volleyball players.” Whilst this statement does allude to a purpose. It would be terrific if the authors could hypothesise a number of potential outcomes. For example, how do they think salivary alpha-amylase, cortisol, and progesterone will behave in the respective teams?

Materials and methods
Whilst the data obtained from such a study is very interesting. The age, sex, and number of participants greatly reduces the generalisability of any findings – as the research team are likely aware. Concerning statistical analysis, the authors state that, “The normality distribution of data was checked with The Shapiro-Wilks test. A paired t-test analysis was applied to compare the mean of variables before and after the match. Pearson's correlation test was performed to evaluate the association between psychological and hormonal parameters.” The Shapiro-Wilks test is the appropriate test to use in terms of evaluating the distribution of data (normal or not). However, it would be more transparent if the authors were to include the p-value of this test after the above statement. If indeed, the data is normally distributed, then a parametric test such as the paired t-test (which the authors chose) can be employed. However, it is worth noting that in a small cohort the use of a non-parametric test, such as the Wilcoxon signed-rank test, may be a more prudent choice. This is plainly because the assumptions of the paired t-test may not be met regarding a smaller sample size – the caveat here is that the Wilcoxon signed-rank test carries less statistical power than the paired t-test.

Conclusions
The take-home messages are clearly defined. However, it may be a stretch to say that cortisol and progesterone have an influence on sports performance at large. This study has shown that in a cohort of 20 (datasets analysed) adolescent, female volleyball players, that those who won the game exhibited a significant decrease in salivary cortisol, whilst those who lost also showed a decrease that was not significant. The same trend was observed for progesterone; however, statistical significance was not achieved in either group. Participants in the winning group were less anxious than their losing counterparts, as assessed by the Sports Anxiety Scale-2 (SAS-2) – statistical significance was not achieved. It may be more prudent for the authors to suggest that hormones such as cortisol and progesterone “can” exert an influence on performance, as there is no definitive evidence that either hormone is the cause of winning and/or losing.

I would like to thank the authors for considering all of the comments above and wish them the best of luck with the publishing process.

References
Fuentes-Rodriguez G, Garcia-Lopez LJ, Garcia-Trujillo V. Exploring the role of the DSM-5 performance-only specifier in adolescents with social anxiety disorder. Psychiatry Res. 2018; 270:1033-1038.

Reviewer 2 ·

Basic reporting

The study entitled “Hormonal and genetics influence on performance anxiety in adolescent female volleyball players” is interesting, but in my opinion suffers from major flaws and lacks some focus.
The study can be divided into three parts:
1) Psychological assements – this part should be considered as a starting point of the whole study, but, surprisingly this part is not represented in the title of the manuscript nor in the abstract. Moreover, the results of this part should be presented as the first, since, the readers would like know whether the players indeed feel some anxiety or stress. What is more, the results in Tab 3 are said to present “Pre- and post-match” but it seems to represent only one set data but it is unclear which one it is. This is critical in this part of the manuscript to find out the effect of the match itself as a stress generator.
2) Hormonal evaluation: the rationale for the assessment of these markers is clear, however, it would be beneficial to broaden the spectrum of stress markers (like here: Giacomello et al. “Current methods for stress marker detection in saliva” 2020.) including training load markers. How the salivary flow rate was measured? – methods are missing.
3) Genetical part – this is to a great extent irrelevant. The rationale for choosing these two polymorphisms in the two genes is very vague. What is the relation between BDNF and progesterone? This part of the manuscript is poorly documented (gel photos should be included) and the number of individuals enrolled (n=12) is minimal.
The conclusions in the abstract are way different than the ones in the manuscript, and in both cases are greatly exaggerated. i.e - the progesterone and cortisol level were no different in losers and winners before the match, thus, how can you conclude that “we can state that certain hormones such as cortisol and progesterone certainly have an influence on sports performance”?
How can you assess performance and generalize the effects by evaluating only 12 players in the single match?
What is more, the manuscript should be professionally proofread as it contains multiple grammar errors and awkward sentences. i.e.
“it measure three components of anxiety”
“also the individualism is crucial to know if the subject act in an individualistic way”
“progesterone shows the reasons why a person builds”
„the corpus lutein”
„lutein phase”
“stated the levels of progesterone”
„was not significant statistical”
Lines 152-156 – lacks citations

Experimental design

as indicated above

Validity of the findings

as indicated above

---

## Round 0.2 · Minor Revisions

Dear authors, thank you for your re-submission. Please, address the minor issues as per the reviewer's suggestion. Be consistent in the use of your acronyms. And make sure to proofread (text and figures, incl. references) the entire manuscript.

**Language Note:** The Academic Editor has identified that the English language must be improved. PeerJ can provide language editing services - please contact us at copyediting@peerj.com for pricing (be sure to provide your manuscript number and title). Alternatively, you should make your own arrangements to improve the language quality and provide details in your response letter. – PeerJ Staff

·

Basic reporting

Basic Reporting
At this point it can get tedious – I understand, trust me. I believe that the material contained within this paper is of value, if it can be presented in the right way. The basic reporting element has most certainly improved. However, there are still a number of English language issues to address, which would greatly enhance the reading experience. For examples, and ways to tackle this, have a look at the “General comments – Abstract” section.

Experimental design

Experimental design
Nothing to add from the last review.

Validity of the findings

Validity of the findings
Reporting of results is satisfactory and reasonably easy to follow. Conclusions are adequate and related to supporting results.

Additional comments

General comments
Abstract
The sentence, “This can be influenced by social closeness this aspect is the closeness towards another individual or a group” may be better worded as, “This can be influenced by social closeness, which is a collective bond shared by individuals or groups.”
The sentence, “In this regard, many studies have focused on the exercise response as a function of ovarian hormones and it has been observed that progesterone is a hormone that plays a key role in reducing anxiety, and thus stress, in humans and other animals” may be better worded as, “In this regard, many studies have focused on the exercise response as a function of ovarian hormones, and it has been observed that the hormone progesterone plays a key role in reducing anxiety, and possibly stress, in humans and other animals.”
The sentence, “On the other hand, high cortisol concentrations are known to contribute to increased anxiety levels” may be better worded as, “On the other hand, high cortisol concentrations have been linked to increased anxiety levels.”
The sentence, “However, the alpha-amylase enzyme (a-Amy) has been suggested as marker of acute stress than cortisol” may be better worded as, “However, the alpha-amylase enzyme (AmyL) has been suggested to be a more effective marker of acute stress, in comparison to cortisol.”
The sentence, “Genetics also seem to influence anxiety and stress management as in the case of brain-derived neurotrophic factor (BDNF) and striatal dopamine transporter (DAT)” may be better worded as, “The genetic profile of an individual may also influence anxiety and stress management, as in the case of brain-derived neurotrophic factor (BDNF) and striatal dopamine transporters (DAT).” ** Insert a brief sentence giving an example of how genetics influences anxiety and stress management.
The methods section may be better worded as, “Twenty-six female volleyball players, who took part in a provincial final match, volunteered to participate in this study. All participants were between the follicular and early ovulatory phase of the ovarian cycle.” ** Insert a brief sentence on the methods used to analyse anxiety and stress (i.e., both salivary alpha-amylase and cortisol were employed as markers of stress. The SAS-2 was used to assess self-reported anxiety. Include genetic elements also).
The results section may be better worded as, “A significant decrease in salivary cortisol, from pre- to post-match levels (7.7 ng/mL vs 4.5 ng/mL), was observed in the winning group (p < 0.039). A significant increase in in salivary alpha-amylase, from pre- to post-match levels, was observed in the winning group (insert data and p value).”
The conclusion primarily focuses on the SAS-2. It would be of great benefit to discuss the SAS-2 within the results section, then tell combine this with the salivary markers to formulate a conclusion. For example, your data shows that the losing group was more anxious than the winning group, and that they also had a high cortisol response. That’s an interesting conclusion. Now, as for your alpha amylase data, that is another point of interest, because now you have two alleged markers of stress doing different things – cortisol down for both groups and alpha-amylase up for both. Going back to your abstract, it begs the question is salivary alpha-amylase a better marker than salivary cortisol? Great talking point for your discussion.
One additional comment: I’ve noticed the paper contains a few acronyms for salivary alpha-amylase. Best to stick with the one. SAA, AmyL, Amy, SαA, αA – take your pick.

Introduction
Greatly improved. Have a member of the team act as editor and polish-up writing style. For example, “Furthermore, also the individualism is crucial to know if the subject acts in an individualistic way; the main instrument used to detect this status is the Auckland Individualism and Collectivism Scale” is a difficult sentence to understand.

Materials and methods
Nothing to add from the last review.

Conclusions
Brilliant work – reads much better.

Reviewer 2 ·

Basic reporting

The authors responded to my questions adeqately.

Experimental design

The authors responded to my questions adeqately.

Validity of the findings

The authors responded to my questions adeqately.

---

## Round 0.3 · accepted · Accept

Many thanks for your submission and collaboration. I am happy to let you know that your manuscript is now accepted for publication.